# Algicidal Molecular Mechanism and Toxicological Degradation of *Microcystis aeruginosa* by White-Rot Fungi

**DOI:** 10.3390/toxins12060406

**Published:** 2020-06-19

**Authors:** Guoming Zeng, Pei Gao, Jiale Wang, Jinxi Zhang, Maolan Zhang, Da Sun

**Affiliations:** 1Chongqing Engineering Laboratory of Nano/Micro Biological Medicine Detection Technology, School of Architecture and Engineering, Chongqing University of Science and Technology, Chongqing 401331, China; 2017015@cqust.edu.cn (G.Z.); 2020005@cqust.edu.cn (P.G.); 2019048@cqust.edu.cn (J.W.); zhangjinxi1223@163.com (J.Z.); 2China Metallurgical Construction Engineering Group Co., Ltd., Chongqing 643000, China; 3Institute of Life Sciences & Biomedicine Collaborative Innovation Center, Wenzhou University, Wenzhou 325035, China

**Keywords:** white-rot fungi, *Microcystis aeruginosa*, molecular mechanism, microcystins

## Abstract

Current research on the inhibition of *Microcystis aeruginosa* growth is primarily focused on algae-lysing bacteria, and few studies have investigated the inhibitory mechanisms by which fungi affect it at the molecular level. A comparative analysis of the effects of *Phanerochaete chrysosporium* on the expression of the algal cell antioxidant protease synthesis gene *prx*, the biological macromolecule damage and repair genes *recA*, *grpE*, and *fabZ*, and the photosynthesis system-related genes *psaB*, *psbD1* and *rbcL*, as well as genes for algal toxin synthesis *mcyB*, were performed to elucidate the molecular mechanism of *Phanerochaete chrysosporium* against *Microcystis aeruginosa* cells. RT-qPCR technology was used to study the molecular mechanism of algal cell inhibition by *Phanerochaete chrysosporium* liquid containing metabolites of *Phanerochaete chrysosporium*, *Phanerochaete chrysosporium* supernatant and *Phanerochaete chrysosporium* inactivated via high temperature sterilization at the gene expression level. Compared with the control, the chlorophyll-a contents dropped, and the *recA*, *grpE*, *fabZ*, and *prx* increased, but the *psaB*, *psbD1*, *rbcL* and *mcyB* showed that they were significantly reduced, which indicated that *Phanerochaete chrysosporium* can not only effectively destroy algal cells, but they may also reduce the expression of the *Microcystis aeruginosa* toxin gene and significantly block the metabolic system underlying the growth of algal cells and the synthesis of microcystins.

## 1. Introduction

Eutrophication is currently one of the most serious water pollution problems faced by natural water bodies such as lakes and rivers in the world. It seriously affects the sustainable comprehensive utilization of water resources and causes harm to people’s drinking water, industrial and agricultural water use, breeding and human health [1,2,3,4,5]. Most of the microbial algae control technologies developed to date rely on algae-lysing bacteria [6,7,8], and the existing studies [9,10,11] have indicated that the *Phanerochaete chrysosporium* which control algae may be great in promoting the control of eutrophication by microbial methods. With the development of molecular biology, the molecular mechanism of the life activities of cyanobacteria and *Microcystis aeruginosa* has attracted people’s attention [12,13,14,15]. In recent years, the mechanisms involved in synthesizing various substances in *M. aeruginosa* cells [16,17], especially the molecular mechanisms that inhibit the growth of *M. aeruginosa*, have focused on biological rhythms, toxin synthesis, phycobiliprotein synthesis and its regulation, ATP synthetase [18,19,20,21], and the molecular response mechanisms through which algae removal technology affects the life activities of cyanobacteria, such as the regulation and expression of biorhythms and the regulation of algal toxin synthesis, have attracted increasing attention [22,23,24].

Currently, microalgal lysis refers to the inhibition of growth or killing of algal cells through direct or indirect lysis. There are two primary modes of action [25]; direct lysis, in which the fungus explicitly contacts or invades algal cells; and indirect lysis, where fungi competes with algae for limited nutrients or secretes extracellular algal lyases. However, the algae suppression mechanism of cyanobacteria, especially on changes in the expression of algae-related genes during the algal lysis process, is relatively lacking. Thus, it is of high important to use modern molecular biology methods to reveal the response mechanism of algal cell growth and metabolism, in accordance with the algal photosynthetic systems and biorhythms, and to use this information to develop an efficient and stable white rot fungal control technology for microalgal population control.

To study the algae-using mechanism of white-rot fungi, we chose *Ph. chrysosporium*, one of the typical white-rot fungi; *Ph. chrysosporium* liquid contained metabolites of *Ph. chrysosporium*, and *Ph. chrysosporium* supernatant and *Ph. chrysosporium* were inactivated via high temperature sterilization, as research objects. In this paper, the antioxidant protease synthesis gene (*prx*), biological macromolecule damage and repair genes (*recA*, *grpE* and *fabZ*), photosynthesis system-related genes (*psaB*, *psbD1* and *rbcL*) and the expression of an algal toxin synthase-related gene (*mcyB*) were the target genes, and RT-qPCR experiments were used to analyze the expression of specific genes after treating *M. aeruginosa* with three treatments. The algal cell inhibition by *Ph. chrysosporium* from the gene expression level in the algal cells could reveal the molecular damage mechanism against the algal cells and the algal toxin degradation.

## 2. Results and Discussion

### 2.1. The Algal Inhibition Effect of Ph. chrysosporium

The chlorophyll-a contents of *M. aeruginosa* were measured after treating this algal with *Ph. chrysosporium* liquid contained metabolites of *Ph. chrysosporium*, *Ph. chrysosporium* supernatant and *Ph. chrysosporium* inactivated via high temperature sterilization (Figure 1). The chlorophyll-a contents were all reduced in the three treatments, especially in the *Ph. chrysosporium* liquid, by 88.6 ± 0.52%, 63.1 ± 0.47% and 27.2 ± 0.65%, respectively. To quantify the inhibitory effect of *Ph. chrysosporium* on *M. aeruginosa*, we used cell counting to detect the number of *M. aeruginosa* cells at different time points. We observed that the *Ph. chrysosporium* had a strong inhibitory effect on *M. aeruginosa*, and the inhibitory effect was time-dependent (Figure 2). In addition, the *M. aeruginosa* cells were smaller, as were the chlorophyll-a contents. These results showed that using *Ph. chrysosporium* may be a feasible way to control harmful algal blooms in the future.

### 2.2. PCR Amplification of the 16S rRNA Internal Standard Gene

After the amplification of the internal reference 16S rRNA gene by PCR, the results of the gel electrophoresis and gel imaging were captured and are shown below (Figure 3). Sample 1 represents *M. aeruginosa* cultured without *Ph. chrysosporium*, and 2–4 represents the *M. aeruginosa* co-cultured with *Ph. chrysosporium* liquid, which contained metabolites of *Ph. chrysosporium*, *Ph. chrysosporium* supernatant and *Ph. chrysosporium* inactivated via high temperature sterilization. As shown in Figure 3, the internal control PCR amplification bands are clear, and can be used for subsequent real-time quantitative reactions.

### 2.3. Effect of Ph. chrysosporium Treatment on the Transcription and Expression of Macromolecule Related Genes in Algal Cells

To study whether the Ph. chrysosporium treatment caused damage to biological macromolecules such as DNA, proteins, and lipids in normal algal cells, recA, grpE, fabZ, and prx were selected as target genes in this experiment (Figure 4).

The *recA* gene encodes *recA* protein, which binds to single-stranded DNA to form a DNA-protein filament. The activated *recA* protein activates the latent protease activity of LexA, resulting in the self-cleavage of proteins such as LexA and the initiation of emergency DNA repair [26]. The *grpE* gene encodes the stress protein *grpE*, which belongs to the DnaK-DnaJ-GrpE molecular chaperone complex system. Its function is to prevent proteins from being denatured by environmental stresses such as heat shock. Cell exposure to severe oxidative stress reportedly often leads to the translation and expression of molecular chaperones and protease molecules. The transcription level of the *grpE* gene promoter in *Escherichia coli* has been shown to increase under oxidative stress [27,28]. The *fabZ* gene, which belongs to the fatty acid synthase II system, encodes a β-hydroxylipoyl carrier protein dehydrogenase that effectively catalyzes the dehydrogenation of the short-chain light-fat phthalophore carrier proteins and long-chain saturated and unsaturated light-fat phthalophore carrier proteins. Cell membranes are susceptible to oxidative stress because they contain polyunsaturated fatty acids. The *prx* gene encodes a 25-kDa antioxidant protease (*prx*). The *prx* protein is part of a thiol-type antioxidant enzyme system. It uses thioredoxin and other sulfur-reducing sites as electron acceptors, to catalyze the reduction of H_2_O_2_, alkyl peroxide and peroxynitrite [29]. *Prx* plays an important role in biological cells and is highly conserved among prokaryotes and eukaryotes. A genomic analysis has shown that the Prx-s gene cluster family is present in cyanobacteria and that it has important biological functions [30,31]. After the algal cells were subjected to environmental stressors, the *recA* gene transcription increased. Three independent experiments were carried out in the below experiments, and each experiment was set in parallel.

The measured data in this experiment were processed with Microsoft Excel 2003 and tested for the significance of the difference. Take *p* < 0.05 as the significant difference; *p* > 0.05 has no significant difference. The data presented in Figure 4 showed that the transcriptional expression of macromolecule-related gene levels of *recA*, *grpE, fabZ* and *prx* in the three experimental groups that received *Ph. chrysosporium* treatment were similar and that the relative expression of these genes after each of the treatments differed significantly from that in the control, and the relative expression of all three genes was significantly increased. The relative expression levels of *recA*, *grpE, fabZ* and *prx* in microalgae were 0.50 ± 0.08%, 0.74 ± 0.02%, 0.49 ± 0.08% and 1.12 ± 0.23%, respectively. In the three experimental groups, the *recA* expression increased to 6.58 ± 0.64%, 5.23 ± 0.49%, and 2.37 ± 0.34%, the *grpE* expression increased to 5.51 ± 0.16%, 3.52 ± 0.18%, and 1.43 ± 0.15%, the *fabZ* expression increased to 10.0 ± 0.58%, 3.90 ± 0.50%, and 2.24 ± 0.46%, and the *prx* expression increased to 9.26 ± 0.79%, 5.51 ± 0.70%, and 2.19 ± 0.19%. It was therefore likely that algal DNA was damaged, and emergency repair may have been initiated. The results of this study also indicated that the protein molecules and lipid macromolecules in the algal cells are severely damaged, and the cells increase their expression of stress proteins and lipid production, to prevent damage to protein molecules and the antioxidant enzyme system, and the repair ability of intracellular proteins is weakened or lost.

### 2.4. Effect of Phanerochaete Chrysosporium Treatment on the Transcription and Expression of Photosynthesis-Related Genes in Algal Cells

The *psaB*, *psbD1* and *rbcL* genes are all involved in the photosynthetic process [32]. The *psaB* encodes a subunit protein of the photosynthetic system I response center, *psbD1* encodes the D2 protein of the photosynthetic system II response center, and the rbcL gene encodes the algae A large subunit protein of the cellular Rubisco process. The data presented in Figure 5 showed that the transcriptional expression levels of *psaB*, *psbD1* and *rbcL* in the three experimental groups that received *Ph. chrysosporium* treatment were similar, and that the relative expression of these genes after each of the treatments differed significantly from that in the control, and the relative expression of all three genes was significantly decreased. The relative expression levels of *psaB*, *psbD1* and *rbcL* in microalgae were 10.0 ± 0.42%, 6.40 ± 0.57% and 3.16 ± 0.63%, respectively. In the three experimental groups, the *psaB* expression decreased to 0.16 ± 0.01%, 0.44 ± 0.07%, and 1.0 ± 0.07%, the *psbD1* expression decreased to 0.47 ± 0.06%, 1.48 ± 0.12%, and 2.51 ± 0.47%, and the *rbcL* expression decreased to 0.25 ± 0.04%, 1.78 ± 0.32%, and 2.51 ± 0.29%; thus, the expression levels of the three groups showed significant changes compared to the control group. Decreased transcription and expression in photosynthesis-related genes may block the electron transfer chain and cause a corresponding reduction in reducing power that affects biological carbon fixation. Therefore, the observed reduction in *psaB*, *rbcL* and *psbD1* gene expression after each of the three *Ph. chrysosporium* treatments may be rate-limiting for carbon fixation, therefore potentially reducing the activity of the Rubisco enzyme and decreasing photosynthesis by the algal cells, which may cause the algal cells to die.

### 2.5. Effect of Phanerochaete Chrysosporium Treatment on the Transcription and Expression of Genes Related to Algal Toxin Synthesis

The *mcyB* gene belongs to the *mcyA-J* gene cluster and encodes for the protein responsible for synthesizing microcystin in a variety of cyanobacteria [33,34,35,36]. Figure 6 showed that the relative expression of *mcyB* in control treatments is 5.25 ± 0.62% and that the *mcyB* expression in the three experimental groups changed significantly compared to the control group, as indicated by the expression levels of 0.46 ± 0.07%, 1.95 ± 0.16%, and 4.29 ± 0.31% observed in the experimental groups. The change in the transcription and expression of the *mcyB* gene may result in a change in the amount of microcystin toxin synthesis in the *M. aeruginosa* cells. The results of this study indicated that the expression of the *M. aeruginosa* toxin gen may result in a change by treating with *Ph. chrysosporium* may block the expression of the *M. aeruginosa* toxin gene; thus, it may be considered a safer method of algal suppression.

## 3. Conclusions

Treating *M. aeruginosa* with *Ph. chrysosporium* was shown to alter the gene expression in *M. aeruginosa* cells significantly. Compared with the control group, the antioxidant protease synthesis gene *prx*, the biomacromolecular damage and the repair genes *recA*, *grpE*, and *fabZ*, were increased, but the photosynthesis system-related genes *psaB*, *psbD1* and *rbcL*, and the algal toxin synthase-related gene *mcyB* showed significantly reduced transcriptional expression in the treated cultures. These results indicate that *Ph. chrysosporium* may effectively control the growth of *M. aeruginosa* cells and may block the synthesis of microcystin toxin, and can provide a method of microbiological treatment of algal blooms. However, for reasons of ecological safety, tests and assessments of the effect of eco-toxicity should be carried out before future use.

## 4. Materials and Methods

### 4.1. Algal Strains and Fungal Strains Cultivation

*M. aeruginosa* was provided by the Freshwater Algae Culture Collection at the Chinese Academy of Sciences. All stock and experimental cultures were maintained at 25 °C under a 12:12 h (L:D) cycle at approximately 90 µmol photons m^−2^ s^−1^, as provided by cool white fluorescent lamps to achieve exponential growth. The growth medium for *M. aeruginosa* was BG-11 and 1000 mL of distilled water. *Ph. chrysosporium* was provided by the Center of Industrial Culture Collection, China. The strain was maintained on potato dextrose agar (PDA) plates for five days, stored at 4 °C, and subcultured every month. Batch liquid tests were conducted in 500 mL beakers containing 250 mg (dryweight) of *Ph. chrysosporium*. In order to clarify the molecular mechanism of the algae dissolution of *Ph. chrysosporium*, this study adopted three methods: *Ph. chrysosporium* liquid which contained metabolites of *Ph. chrysosporium*, *Ph. chrysosporium* supernatant and *Ph. chrysosporium* inactivated via high temperature sterilization were added to *M. aeruginosa* cultures for the algae lysis experiments. *M. aeruginosa* was treated by *Ph. chrysosporium* under the optimum conditions of 250 mg^−l^
*Ph. chrysosporium* at DO 7.0 mg^−l^ with 25 °C and 12:12h (light:dark) cycle for 24 h. Samples that were prepared in the same way as the test cultures, but lacking *Ph. chrysosporium*, were used as the controls. All experiments were conducted in triplicate.

### 4.2. Total Chlorophyll-a Content Test

The *M. aeruginosa* density was tracked using a counting box and the total chlorophyll-a was extracted with 90% acetone and determined according to the repeated freezing and thawing-extraction method [37].

### 4.3. RNA Extraction

RNA was extracted by the Trizol method, and the steps were as follows:Appropriate amounts of algae tissue were homogenized by grinding in liquid nitrogen for a short time while keeping liquid nitrogen in the mortar (the tissue sample volume did not exceed 10% of the PL volume).The sample was transferred to a 1.5 mL RNase-free centrifuge tube, and 1 mL of PL lysate was added. The sample was mixed by inversion and incubated at 65 °C for 5 min, to decompose the ribosomes completely.The sample was centrifuged at 12,000 rpm for 10 min at room temperature. The resulting supernatant was carefully transferred to a new RNase-free filter column (if floating matter was present on the surface of the supernatant, a pipette tip was used to separate and draw off the liquid below the surface).The filter column was centrifuged at 10,000 rpm for 45 s, and the filtrate (containing the total RNA) was collected in a collection tube.One volume of 70% ethanol was added to the collection tube, after first checking whether absolute ethanol had been added, and the sample was mixed by inversion (precipitation sometimes occurred at this stage). The resulting solution and the precipitate, if any, were transferred to an RA adsorption column. If the volume of solution was large, it was passed through the column in sections. The adsorption column can hold up to 700 μL of solution, and the adsorption column was sleeved in a collection tube.The adsorption column was centrifuged at 10,000 rpm for 45 s; the waste solution was discarded, and the column was returned to the collection tube.After checking whether absolute ethanol had been added to the rinsing solution, 500 μL of RW rinsing solution was added to the column, the column was centrifuged at 12,000 rpm for 60 s, and the waste solution was discarded.The RA adsorption column was returned to the empty collection tube and centrifuged at 12,000 rpm for 2 min, to remove as much of the rinsing solution as possible. This step is necessary because the residual ethanol in the rinsing solution inhibits the enzyme digestion reaction.Optionally, 30 μL of digestion solution was placed in the center of the adsorption membrane, and the sample was incubated at 37 °C for 15–30 min. The digestion solution consisted of 2 μL of RNase-free DNase, 3 μL of DNase 10× Reaction Buffer, and 25 μL of RNase-free water. The amount of RNase-free DNase was adjusted according to the amount of DNA; 1 μL of RNase-free DNase can digest 1 μg of RNA. The amount of reaction buffer was increased or decreased proportionally.Protein-removing solution RE (500 μL) was added to the adsorption membrane; following an incubation at room temperature for 2 min, the column was centrifuged at 12,000 rpm for 45 s, and the waste solution was discarded.After we checked if absolute ethanol had been added to the RW rinsing solution, 500 μL of rinsing solution was added, the sample was centrifuged at 12,000 rpm for 60 s, and the waste solution was discarded.Step 11 was repeated.The RA adsorption column was replaced in the empty collection tube and centrifuged at 12,000 rpm for 2 min, to remove as much of the rinsing solution as possible; this step was necessary to avoid having residual ethanol in the rinsing solution, which would inhibit the downstream reaction.The RA adsorption column was placed in an RNase-free centrifuge tube, and 50–80 μL of RNase-free water that had been heated to 65–70 °C in a water bath was placed in the center of the adsorption membrane. The volume of water used for this step was adjusted according to the expected RNA yield. The column was allowed to remain at room temperature for 2 min and was then centrifuged at 12,000 rpm for 1 min.

### 4.4. RNA Concentration Determination

An Eppendorf BioPhotometer D30 Nucleic Acid Protein Analyzer was used to determine the RNA concentration, based on the A260/A280 ratio.

### 4.5. Reverse Transcription PCR of Algal Cell mRNA


Genomic DNA was removed by incubating the RNA sample (total RNA 700 ng) with 4.0 μL of 5× gDNA Eraser Buffer, 2.0 μL of gDNA Eraser, and RNase-free dH_2_O, in a total volume of 20 μL. The reaction system was incubated at 40 °C for 2 min and then stored at 4 °C in preparation for step 2.cDNA strand synthesis
a)To a 0.2-mL EP tube, 20 μL of the reaction solution from step 1, 2 μL of primer, 2 μL of RNA Eraser, 8 μL of 5× RNA buffer, and 8 μL of RNase-free H_2_O were added, and the sample was mixed slightly.b)The sample was incubated sequentially at 25 °C for 5 min, 37 °C for 20 min, and 85 °C for 1 min. It was then stored at 4 °C for future use or at −20 °C for long-term storage.


### 4.6. PCR Amplification of the 16S rRNA Internal Standard Gene

The 16S rRNA internal standard gene amplification reaction was performed on the extracted cDNA, to measure the efficiency of the reverse transcription reaction [38]. The primer used in this amplification reaction was the gene-specific primer 16S rrn. The reaction consisted of 2 μL of cDNA solution, 1 μL of Primer16s (fo) (re) (10×), 5 μL of PCR buffer II (10×), 2 μL of dNTP mix (10 mM), 0.5 μL of TaKaRa Ex Taq HS (5 U/μL), and 39.5 μL of RNase-free H2O (total volume, 50 μL). The PCR reaction program was 95 °C for 30 s, followed by 40 cycles of 95 °C for 5 s and 60 °C for 60 s.

### 4.7. Agarose Gel Electrophoresis

The amount of agar powder necessary to produce a 1% agarose gel was placed in a conical flask, and an appropriate amount of TAE running buffer was added. The flask was microwaved to completely dissolve the agar powder and make the solution transparent, and it was shaken slightly to obtain a glue. When the solution reached approximately 60 °C, an appropriate amount of 4 s GelRed nucleic acid dye was added.The gel mold was placed horizontally, a comb was inserted at one end, and agarose solution at a temperature of approximately 60 °C was slowly poured into the groove, to form a uniform horizontal gel surface.When the gel had solidified, the comb was carefully removed by pulling upward, so that the cathode section at the end of the loading hole was placed in the electrophoresis tank.TAE running buffer was added to the tank until the liquid covered the gel surface.When the PCR reaction was complete, the samples were mixed with loading buffer in a suitable ratio and loaded into the gel wells, using a pipette.The power to the electrophoresis apparatus was turned on, and the voltage was adjusted to 100 V to stabilize the output.The position of the blue band formed by the bromophenol dye was observed. When it had traversed approximately 2/3 of the gel length, the electrophoresis was stopped.A piece of plastic wrap was placed on the sample table of the UV fluorometer, the air bubbles were removed, and the gel was placed on the surface. The outer door of the sample chamber was closed, the UV illumination (365 nm) was turned on, and the gel was observed through the observation port.

### 4.8. Real-Time PCR Experiment

The primers used in Table 1 were completed by the Shanghai Shengong Biological Engineering Company (Shanghai, China).

Amplification was performed in a Bio-Rad CFX Amplifier. The amplification system consisted of 5 μL of SYBR, 0.5 μL of primer (fo) (re) (10×), 1 μL of cDNA, and 3 μL of RNase-free H_2_O, in a total volume of 10 μL.

A two-step PCR amplification program was used. For the first stage (pre-denaturation), the sample was incubated at 95 °C for 3 min. For the second stage, the PCR consisted of 40 cycles of 95 °C for 10 s and 60 °C for 30 s. The expression level of the related group mRNA is expressed by measuring the Ct value, and 16S rRNA is used as an internal reference. The formula used to calculate the relative expression of the target genes was as follows:2-ΔΔCt = 2^−[(Cttarget-CtHousekeeping)test.(Cttarget-CtHousekeeping)control]^(1)
where Ct^target^ is the average Ct value of the target gene, Ct^Housekeeping^ is the average Ct value of the housekeeping gene, (Ct^target^-Ct^Housekeeping^) test is the relative expression of the test group, (Ct^target^-Ct^Housekeeping^) control is the relative expression of the control group, and 2-ΔΔCt represents the relative expression of the target gene. The results were collated, counted, and plotted using Origin 8.0 (Originlab, Northampton, MA, USA). The data are expressed as means ± SD (*n* = 3).

### 4.9. Analytical Method

Each of the above experiments was conducted independently, and each experiment was set in parallel. The measured data in this experiment were used in Origin 8.0 and tested for significance of differences. With *p* < 0.05 as the significant difference, *p* > 0.05 has no significant difference. The data are expressed as the means ± SD (*n* = 3).

## Figures and Tables

**Figure 1 toxins-12-00406-f001:**
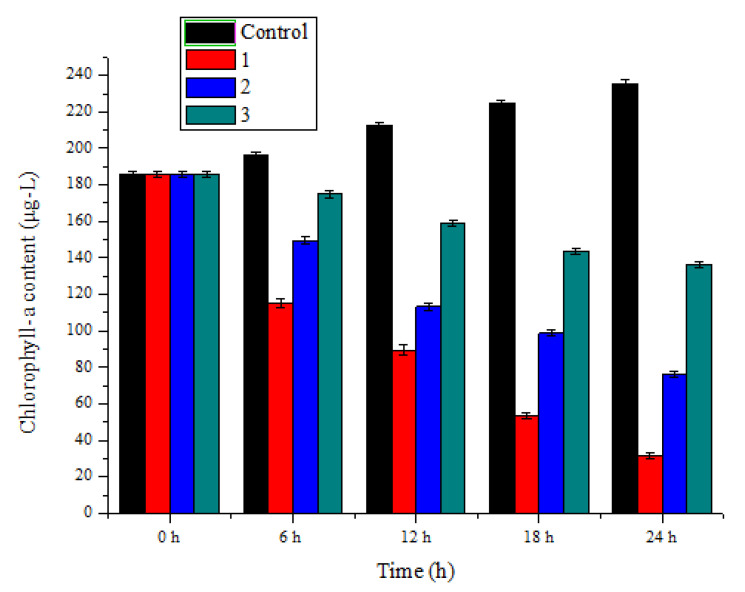
Algicidal efficiency of *M. aeruginosa* treated by *Ph. chrysosporium*. 1—presents the content of chlorophyll-a in the culture of *M. aeruginosa* with *Ph. chrysosporium* liquid which contained metabolites of *Ph. chrysosporium*; 2—presents content of chlorophyll-a in a *M. aeruginosa* culture *Ph. chrysosporium* supernatant; 3—presents content of chlorophyll-a in a *M. aeruginosa* culture with *Ph. chrysosporium* inactivated by high temperature. *M. aeruginosa* Data are expressed as mean ± SD (*n* = 3).

**Figure 2 toxins-12-00406-f002:**
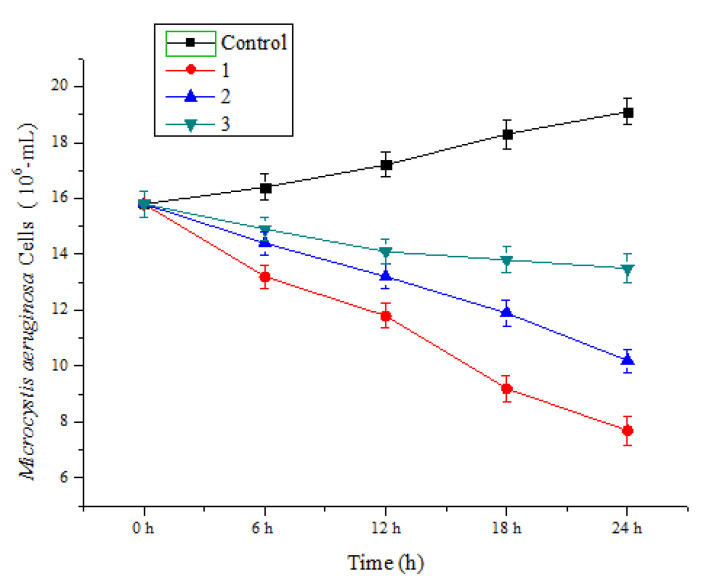
*M. aeruginosa* Cells treated by *Ph. chrysosporium*. 1—presents the culture of *M. aeruginosa* with *Ph. chrysosporium* liquid which contained metabolites of *Ph. chrysosporium*; 2—presents the *M. aeruginosa* culture *Ph. chrysosporium* supernatant; 3—presents the *M. aeruginosa* culture with *Ph. chrysosporium* inactivated by high temperature. Data are expressed as mean ± SD (*n* = 3).

**Figure 3 toxins-12-00406-f003:**
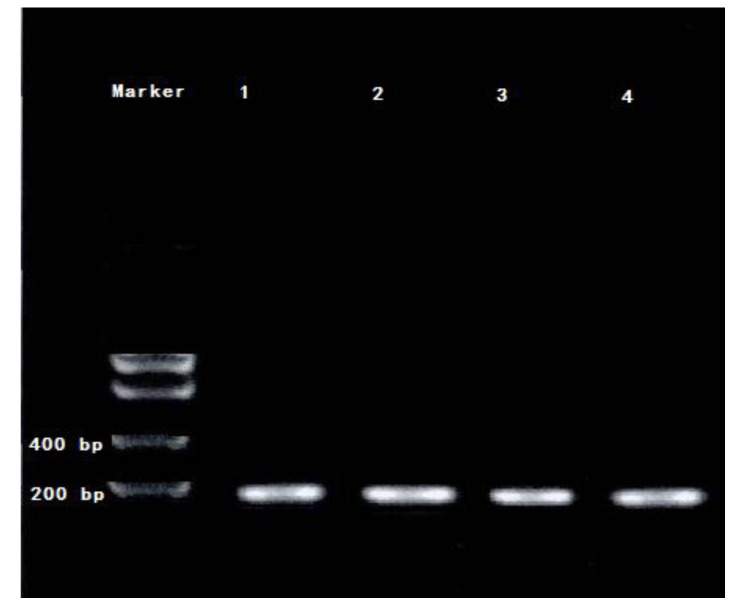
The PCR amplification electrophoregram of 16S rRNA, 1 represented the sample was *M. aeruginosa* cultured without *Ph. chrysosporium*, and 2—presents the *Ph. chrysosporium* liquid which contained metabolites of *Ph. chrysosporium*; 3—presents the *M. aeruginosa* culture *Ph. chrysosporium* supernatant; 4—presents the *M. aeruginos*a culture with *Ph. chrysosporium* inactivated by high temperature.

**Figure 4 toxins-12-00406-f004:**
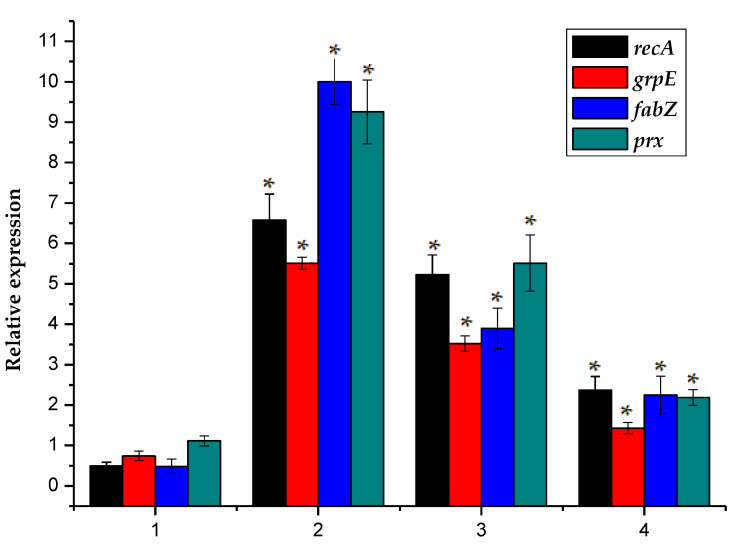
Here, 1 represented the sample *M. aeruginosa* cultured without *Phanerochaete Ph. chrysosporium*, and 2–4 represented the normalized relative expression of *recA, grpE, fabZ, and prx* gene in the *M. aeruginosa* co-cultured with *Phanerochaete Ph. chrysosporium* liquid, which contained metabolites of *Phanerochaete Ph. chrysosporium*, *Phanerochaete Ph. chrysosporium* supernatant and *Phanerochaete Ph. chrysosporium* inactivated via high temperature sterilization. The error bar is the mean ± standard deviation. * is the *p* < 0.05.

**Figure 5 toxins-12-00406-f005:**
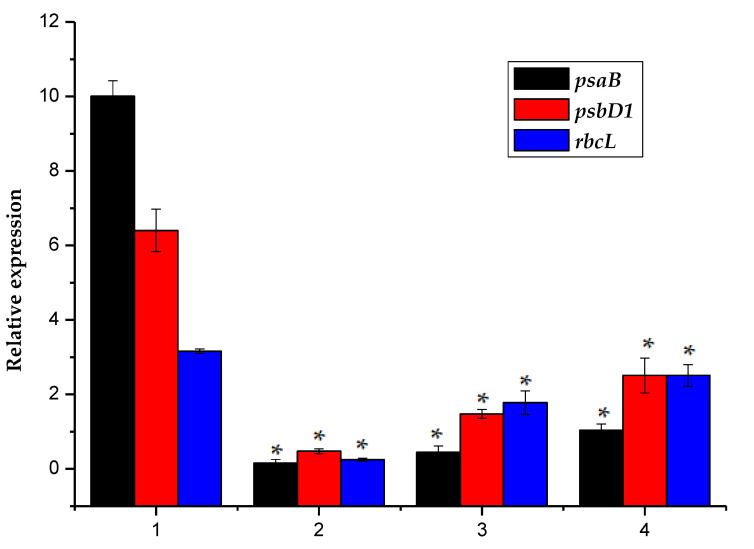
Here, 1 represented the sample *M. aeruginosa*, cultured without *Phanerochaete Ph. chrysosporium*, and 2–4 represented the normalized relative expression of *psaB, psbD1, and rbcL* gene in the *M. aeruginosa* co-cultured with *Phanerochaete Ph. chrysosporium* liquid, which contained metabolites of *Phanerochaete Ph. chrysosporium*, *Phanerochaete Ph. chrysosporium* supernatant and *Phanerochaete Ph. chrysosporium* inactivated via high temperature sterilization. The error bar is the mean ± standard deviation. * is the *p* < 0.05.

**Figure 6 toxins-12-00406-f006:**
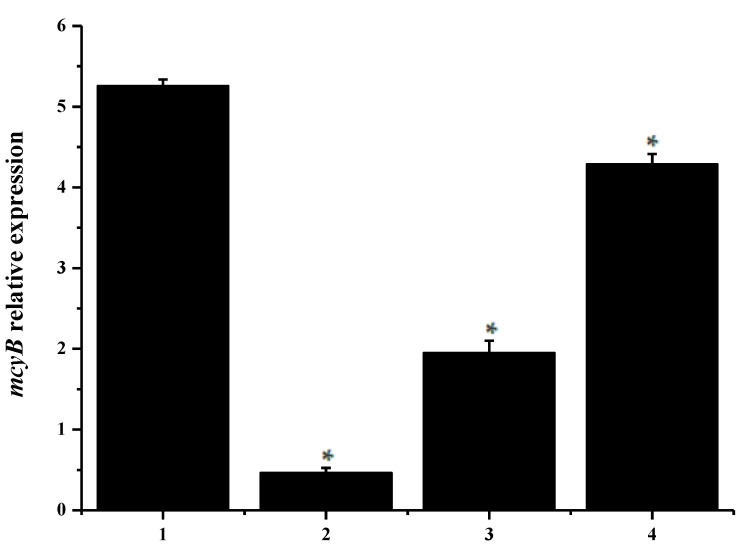
Here, 1 represented the sample *M. aeruginosa* cultured without *Ph. chrysosporium*, and 2–4 represented the normalized relative expression of *mcyB* gene in the *M. aeruginosa* co-cultured with *Ph. chrysosporium* liquid, which contained metabolites of *Ph. chrysosporium*, *Ph. chrysosporium* supernatant and *Ph. chrysosporium* inactivated via high temperature sterilization. The error bar is the mean ± standard deviation. * is the *p* < 0.05.

**Table 1 toxins-12-00406-t001:** The primer sequences.

Primers	Sequences (5’ to 3’)
16S rrn-Fo (forward)	GGACGGGTGAGTAACGCGTA
16S rrn-Re (reverse)	CCCATTGCGGAAAATTCCCC
*prx*-Fo	GCGAATTTAGCAGTATCAACACC
*prx*-Re	GCGGTGCTGA TTTCTTTTTTC
*mcyB*-Fo	CCTACCGAGCGCTTGGG
*mcyB*-Re	GAAAATCCCCTAAAGATTCCTGAGT
*recA*-Fo	TAGTTGACCAGTTAGTGCGTTCTT
*recA*-Re	CACTTCAGGATTGCCGTAGGT
*grpE*-Fo	CGCAAACGCACAGCCAAGGAA
*grpE*-Re	GTGAATACCCATCTCGCCATC
*fabZ*-Fo	TGTTAATTGTGGAATCCATGG
*fabZ*-Re	TTGCTTCCCCTTGCATTTT
*psbD1*-Fo	TCTTCGGCATCGCTTTCTC
*psbD1*-Re	CACCCACAGCACTCATCCA
*psaB*-Fo	CGGTGACTGGGGTGTGTATG
*psaB*-Re	ACTCGGTTTGGGGATGGA
*rbcL*-Fo	CGTTTCCCCGTCGCTTT
rbcL-Re	CCGAGTTTGGGTTTGATGGT

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
