# Peer review of "Algicidal Molecular Mechanism and Toxicological Degradation of Microcystis aeruginosa by White-Rot Fungi"

_toxins, 2020, doi:10.3390/toxins12060406_

Round 1

Reviewer 1 Report

This is a much better version, but still require correction.

Line 22: after blooms need a space.

Line 24: White not white

Line 25-27: Better to write in two different sentences.

Line 31: possess eutrophication ( no article before eutrophication)

Line 36: that addressed

Line 36 later part statement requires reference.

Line 40: Write down 2 days (d) and then next time use d

Line 60-65: please use past from ( were not are)

Line 74: 885.91±0.52% is it right???

Figure 1 and Figure 2 grammar are not correct, please make correction (1-3 represented  the was ). Did you miss sample???

Figure 3 require size of marker

Line 250: This section requires more information about the preparation of three treatments.

Line 233:grpE and psaB gene didn’t show significantly reduced transcriptional expression in the treatment of inactivate white rot fungus. ( you need to mention this in conclusion and need to explain why in the discussion)

Author Response

Dear Editor and reviewer,

Here enclosed is a completely new manuscript entitled “Algicidal Molecular Mechanism and Toxicological Degradation of Microcystis aeruginosa by white-rot fungi”, which we wish to be considered. The molecular algicidal mechanism of Microcystis aeruginosa by Phanerochaete chrysosporium was studied for the first time,and we think It is of great significance in promoting the control of eutrophication by microbial methods.We believe that this new paper may be also of particular interest to the readers of your journal.
Thanks very much for your attention to our paper.
Warm regards,
The authors

Reviewer 2 Report

The text submitted for review discusses the results of studies on the molecular lysing mechanism of the white rot fungus, which were carried out on a group of genes responsible for the synthesis of antioxidant protease (prx), the biological macromolecule damage and repair genes (recA, grpE and fabZ), genes associated with the photosynthesis system (piesB , psbD 1 and rbcL) and the gene associated with miccrocystin synthase (mcyB). The results of the analysis of the expression of these genes after the cultivation of Cyanobacteria Microcystis aeruginosa with the fungus, with the inactivated fungus and with its metabolites have enabled the demonstration of the strong interaction of this fungus at many levels of M. aeruginosa metabilism. For this reason, I find the work interesting and deserving of publication. However, numerous remarks force me to recommend this text for printing only after a significant revision.

Remark (1): No spaces in the expression: „blooms,but” (line 22).

Remark (2):  In the lines 26-27 instead of: "[...] for the first time, and Phanerochaete chrysosporium may effectively remove microcystins" I suggest: "[...] for the first time. It has been shown that Phanerochaete chrysosporium can effectively remove microcystins."

Remark (3): In the first sentence of the Introduction chapter (lines 30-31), the authors confuse causes and effects of eutrophication. Meanwhile, the first two sentences of the abstract in the quoted publication [2] state that the real effect of "algal blooms, including those containing cyanobacteria, are of environmental concern due to the toxicities of some of the constituent microorganisms. This compromises the safety of freshwater causing illness in livestock and humans. " (Smith, J.G .; Daniels V. Algal blooms of the 18th and 19th centuries. Toxicon 2018, 142, 42-44.). In contrast, the next two sentences clearly emphasize that the formation of algal blooms is a consequence of intensive agriculture, which provided nutrients flowing into the water (i.e. eutrophication of water bodies) to "for promoting harmful algal blooms" - the formation of algal blooms is therefore the result of eutrophication of the water, not its cause!).

Remark (4): The second sentence in the chapter "Introduction" is incomplete - there is no statement that the listed phenomena and mechanisms were the subject of research.

Remark (5): Describing the state of research the authors did did not include the  publications of Zeng, Guoming & Wang, Pu & Wang, Ying. (2015). Algicidal efficiency and mechanism of Phanerochaete chrysosporium against harmful algal bloom species. Algal Research. 12. 182-190. 10.1016/j.algal.2015.08.019., in which it was shown that the algal cells of Cryptomonas obovate, Oscillatoria sp. (just like Microcystis aeruginosa, it belongs to Cyanoprokaryota), and Scenedesmus quadricauda  were severely damaged by P. chrysosporium and were finally degraded directly by the fungus, which has good algicidal properties. This article is therefore about the species that is the subject of the publication.

Remark (6):  No spaces: “lysis,where” (line 56).

Remark (7): In the signature of Figure 1, 2 and …. , I suggest clearly assigning the numbers indicating the three treatments mentioned in the text, e.g. „1 - presents the content of chlorophyll-a in the culture of Microcystis aeruginosa with white rot fungus liquid which contained metabolites of white rot fungi; 2 - content of chlorophyll-a in a M. aeruginosa culture with white rot fungus inactivated by high temperature; 3 - content of chlorophyll-a in a M. aeruginosa culture white rot fungus supernatant.”

In addition,  the unit on the Y axis in Figure 1 should be corrected from "ug-l" to "µg-L". The axis descriptions in Figure 2 lack spaces before the parentheses containing the units.

Remark (8): According to Figure 1, the chlorophyll-a contents were all reduced in the first case not by 885.91% (line 74) but at most by 88.6%. I propose, because of the low number of repetitions (n ​​= 3), to give up exaggerated accuracy.

Remark (9): The general rule is that in publications the name of the genre is given in full only once. Then, if the abbreviation is unambiguous, only the first letter of the genus name with a full stop and the species name n are used. In this case, it should be written (apart from the beginning of the sentence and the captions of the drawings): M. aeruginosa. (lines: 38, 64, 70, 75, 76, 78, e.t.c.).

Remark (10): In line 78 "cells were smaller", not "cells were also decreased".

Remark (11): Please correct the description in Figure 3 in accordance with remarks (7) and (9).

Remark (12): The description of the drawing (Figures 4, 5, 6, 7, 8, 9, 10 and 11) should be corrected. First you should indicate what the drawing shows. Poandto, in the situation where the variants of experience are given in the figure, it makes no sense to repeat these information in the signature, especially since the symbols 1-4 are not even marked in the drawing. In addition, there is no units (%) in the axis descriptions in the drawings. The abbreviations "recA", "GrpE", e.t.c. should also be removed from under the x axis, because as the description of the drawing indicates, the expression of these genes is presented on the y axis.

Remark (13): I do not see agreement between the content of the sentence in lines 115 and 116 with Figure 4. According to the figure, the highest gene expression with a value corresponding to 6.58 was found in the control sample and not in the culture with the white rot fungus solution, as stated in the sentence from lines 115-116.

Remark (14): In the sentence from line 131 I suggest removing the statement "Under the effect of this method".

Remark (15): In line 161 has no spaces betwenn “peroxynitrite[31]”,  in line 168 – between the sentences ("was reduced.Same") and dot at the end of the paragraph. “peroxynitrite[31]”.

Remark (16): In line 217 the term "protozoa" is incorrectly used to refer to Cyanobacteria (bleu-alge, Cynophyta). Protozoa are organisms that make up another kingdom.

Remark (17): In line 243 has no spaces betwenn “Sciences.All”.

Remark (18): The description of the experiment on Microsystis aeruginosa contained in lines 257-261 should not be given in the subsection "Fungal strains" but either in a separate subsection or in the previous subsection, which bears (more relevant to this content) the title "Algal strains and cultivation".

Remark (19): In the section "Reference", spaces (required by the journal) are missing in many items.

Author Response

(The authors gave the same response as above.)

Reviewer 3 Report

Dear Authors, 

Please find a detailed breakdown of my comments surrounding your resubmission attached. 

In short, where I can see that changes have been made in places, they are still not to what I would class as publication standard. 

I recommend thorough adoption of the suggested changes before possible resubmission.

Author Response

(The authors gave the same response as above.)

Round 2

Reviewer 2 Report

The topic and research are interesting but I am afraid that the part Introduction, Results and Literature are not final manuscript ready to be published. In my opinion, the concept of the text should be thoroughly considered, i.e. the description of the state of the art in the field of research and the presentation of results, including in particular the number of drawings presented. Even a cursory search of the Internet shows that the authors did not sufficiently query the literature devoted to similar research, which would undoubtedly have an impact on the discussion of the results, but also on their presentation in line with the magazine's standards. Among the unused publications one can cite at least a few titles related to the study of gene expression in Microcystis aeruginosa under the influence of various pressure factors, e.g .:

Jie Wang, Qi Liu, Jia Feng, Jun-ping Lv, Shu-lian Xie, Effect of high-doses pyrogallol on oxidative damage, transcriptional responses and microcystins synthesis in Microcystis aeruginosa TY001 (Cyanobacteria), Ecotoxicology and Environmental Safety, Volume 134, Part 1, 2016, Pages 273-279,https://doi.org/10.1016/j.ecoenv.2016.09.010.

Shao, J., Yu, G., Wang, Z. et al. Towards clarification of the inhibitory mechanism of wheat bran leachate on Microcystis aeruginosa NIES-843 (cyanobacteria): physiological responses. Ecotoxicology 19, 1634–1641 (2010). https://doi.org/10.1007/s10646-010-0549-1

but also:

J-Q Shi, Z-X Wu & L-R Song (2013) Physiological and molecular responses to calcium supplementation in Microcystis aeruginosa (Cyanobacteria), New Zealand Journal of Marine and Freshwater Research, 47:1, 51-61, DOI: 10.1080/00288330.2012.741067

Shao, J., Jiang, Y., Wang, Z. et al. Interactions between algicidal bacteria and the cyanobacterium Microcystis aeruginosa: lytic characteristics and physiological responses in the cyanobacteria. Int. J. Environ. Sci. Technol. 11, 469–476 (2014). https://doi.org/10.1007/s13762-013-0205-4

In my opinion, the meaning of the amended first sentence of the chapter "Introduction" in verses 27-28 is still controversial: "Microcystis aeruginosa [1-6], [....], Which is the main algae species for water eutrophication [7-9]. " In addition, I doubt the validity of quoting three publications in this sentence, which:

- It is not associated with Microcystis aeruginosa (as e.g. cited item No. 7 (Wang, Y., Gong, Y., Dai, L., Sommerfeld, M., Zhang, C., & Hu, Q. (2018). Identification of harmful protozoa in outdoor cultivation of Chlorella and the use of ultrasonication to control contamination. Algal Research, 31, 298-310) in which identified a range of contaminating organisms (2 fungi, 7 flagellates, 3 amoebae, 4 ciliates, 1 rotifer, and 2 large insects) in Chlorella cultures cultivated in outdoor raceway ponds,

- Whereas the second publication cited (No. 8 in the list - Jia, P.L .; Zhou, Y.P .; Zhang, X.F .; Zhang, Y .; Dai, R.H. Cyanobacterium removal and control of algal organic matter (AOM) release by UV / H2O2 pre-oxidationenhanced Fe (II) coagulation. Water res. 2018, 131, 122-130) describes the application of "a pre-oxidation process to assist the subsequent Fe (II) -coagulation-sedimentation process to remove M. aeruginosa and AOM". In the chapter "Introduction" there is a sentence touching the problem of the occurrence of M. aeruginosa in eutrophic waters: "Microcystis aeruginosa (M. aeruginosa), one of the prominent and ubiquitous cyanobacterial species, is the chief culprit of harmful blooms in aquatic environments with eutrophication." This sentence containing the thought cited from the publication Lapointe et al. (2015).

- Finally the third publication cited (No. 9 - Lin, JL; Hua, LC; Wu, YT; Huang, CP Pretreatment of algae-laden and manganese-containing waters by oxidation-assisted coagulation: Effects of oxidation on algal cell viability and manganese precipitation. Water Res. 2016, 89, 261-269) is devoted to studying  "the effects of preoxidation on the performance of coagulation-sedimentation for the simultaneous removal of algae and soluble Mn, including ionic and complexed Mn."

To sum up, an analysis of how to cite literature in this small fragment of the reviewed text does not confirm the behavior of the standards of the journal.

Still visible problems with the proper description of the charts in Figs. 4-11 clearly indicate that the most appropriate would be to properly group them and present these results in a maximum of 2-3 figures thematically related to each other (e.g. Fig. 4. Relative normalized expressions of three genes (recA, grpE, and fabZ) of M. aeruginosa under the stress of ....). It seems that the best would be to group the charts according to the principle of Conclusion (increased and reduced expression)

References at the end of the text are not in the correct format, e.g.:

  1. Jia,-P.L.;-Zhou,-Y.P.;-Zhang,-X.F.;-Zhang,-Y.;-Dai,-R.H.-Cyanobacterium removal and control of algal organic matter (AOM) release by UV/H2O2 pre-oxidationenhanced Fe(II) coagulation.-Water Res. 2018, 131, 122-130.

in eight places (marked in yellow) should be changed.

Other remarks:

- I do not understand the mention of six publications behind the name of the species Microcystis aeruginosa (line 27).

- Fig. 2 as described in lines 67-70 presents changes in the number of M. aeruginosa cells, which is also confirmed by the description of the Y (vertical) axis. Meanwhile, in the description of the symbols 1-3 in the drawing caption, it was stated that they present changes in chlorophyll-a content in various M. aeruginosa cultures. An analogous discrepancy is in the description of Figure 3. It presents the electrophoregram of 16S rRNA, but symbols 1-3 already contain chlorophyll-a concentration.

- The principle of writing Latin names (first for the first time in the text, then most often abbreviated), mentioned in my first review of the text, refers not only to Microcystis aeruginosa, but also to the name of the fungus Phanerochaete chrysosporium. Therefore, in many places of the text instead of the full name should be Ph. chrysosporium, especially in such paragraphs as in lines 49-59, in which the name of this mushroom appears 6 times, and in the paragraph in lines 62-72 - 8 times, etc. However, the rule regarding abbreviations also applies - the full name should start the first paragraph and when the species name appears in the summary and in the captions of the drawings or tables.

Author Response

Dear Reviewer, Here enclosed is a completely new manuscript entitled “Algicidal Molecular Mechanism and Toxicological Degradation of Microcystis aeruginosa by white-rot fungi”, which we wish to be considered. Current research on the inhibition of Microcystis aeruginosa growth is primarily focused on algae-lysing bacteria, and few studies have investigated the inhibitory mechanisms by which fungi affect at the molecular level. A comparative analysis of the effects of Phanerochaete chrysosporium on the expression of the algal cell antioxidant protease synthesis gene prx, the biological macromolecule damage and repair genes recA, grpE, and fabZ, and the photosynthesis system-related genes psaB, psbD1 and rbcL as well as genes for algal toxin synthesis mcyB were performed to elucidate the molecular mechanism of Phanerochaete chrysosporium against Microcystis aeruginosa cells. We believe that this new paper may be also of particular interest to the readers of your journal.

Warm regards

Reviewer 3 Report

Many thanks for implementing the suggested changes. Once you include the appropriate information about your statistical testing and the associated values ie. not just the p value, I will happily recommend you paper for publication.

Kind Regards. 

Author Response

(The authors gave the same response as above.)

Round 3

Reviewer 2 Report

The changes introduced by the authors slightly improved clarity of results presentation. The authors introduced table 2 in which they summarized all results (mean values ​​of relative gene expression and standard deviation). At the same time, they left all charts with this data. What's more, all the values ​​given in Table 2 are additionally repeated in identical form in the text (lines 131, 141-143, 154, 164-166, 247-250). (If there is table 2 in the text, then one should give up literal repetition of these values ​​and remove all drawings from number 4). The authors rejected my proposal to combine Figures 4-7 and 8-11, although they are all referenced once, and some of them together (I quote: "The data presented in Figures 8, 9, and 10" in line 242). In addition, they are all properly identically described, without a clear indication (before mentioning the meaning of the symbols), what is shown in these charts. The descriptions differ only in the name of the gene, as evidenced by the fact that the following descriptions have copied errors (no spaces in the names of species in the same places, and double dots at the end of the sentence).

The current version of the first sentence is also unacceptable. I have consistently asked for an indication that M. aeruginosa the characteristic species of eutrophic waters, not eutrophication (e.g.: Cyanobacteria, especially the Microcystis aeruginosa which is the characteristic species of eutrophic waters, is a major algal species that occurs in lakes, reservoirs, and other water ecosystems in China  [1-5]).

I do not understand the wording used in the description of the new equations in lines 71-76: "regression [...] between M. aeruginosa algal fluid and its algal cell concentration". What do these equations describe? Do changes in gene expression from Figure 2? The authors also did not indicate what the variable x in these equations means.

In line 176, the authors write "The prx expression decreased ....", and in "Conclusion": "Compared with the control group, the antioxidant protease synthesis gene prx, the biomacromolecular damage and repair genes recA, grpE, and fabZ, were increased .... ".

In the last two sentences from verses 143-146, a similar narrative style should be adopted as in the summary of other results (e.g. "results indicate ...").

In addition, the authors did not actually make the effort to carefully review the text and remove numerous minor errors (including in the reference list) - no spaces, double punctuation, etc. Despite the fact that I have paid attention to this issue several times.

Author Response

Dear reviewer,

Thank you for your good comments and suggestions. We have modified figures and some paragraph in my paper,please check the revised paper, thank you.

Warm regards
